# The Role of Selenium and Hydrocarbons in Au-Ag Ore Formation in the Rodnikovoe Low-Sulfidation (LS) Epithermal Deposit, Kamchatka Peninsula, Russia

Nadezhda Tolstykh [1,*], Maria Shapovalova [1], Elena Shaparenko [1] and Daria Bukhanova [2]

[1] Sobolev Institute of Geology and Mineralogy, Siberian Branch of the Russian Academy of Sciences, 630090 Novosibirsk, Russia

[2] Institute of Volcanology and Seismology, Far Eastern Branch of the Russian Academy of Sciences, 683023 Petropavlovsk-Kamchatsky, Russia

\* Correspondence: tolst@igm.nsc.ru

**Abstract:** Gold-silver mineralization in the Rodnikovoe LS epithermal deposit is characterized by selenium speciation. Two main alternating ore assemblages have been identified: silver-aguilarite-acanthite and gold-uytenbogaardtite-acanthite. The former mineral association is intergrown with secondary silver ($Ag_{0.77-0.91}$), whereas the latter assemblage is closely associated with high-grade gold ($Au_{0.63-0.67}$). However, both are dominated by $Ag_{0.49-0.56}Au_{0.44-0.51}$ alloys. The geochemical evolution of the ore-forming system developed in the direction: Fe → Cu; Ag → Au; S → Se; As → Sb. Organic compounds (1 relative %) of both biogenic and thermogenic origin were found in fluid inclusions. These molecules participated in the formation of Ag,Au-complexes and transported noble metals along with selenium. Hydrothermal fluids are characterized by $fSe_2/fS_2$ ratios < 1, conditions such that the deposition of selenide minerals is inhibited, except for the naumannite and acanthite series. These conditions allow active entry of selenium into sulfosalts (the selenium substitutes for sulfur).

**Keywords:** low-sulfidation epithermal deposits; gold-silver ores; selenium; hydrothermal fluids; mineral assemblages; hydrocarbons; acanthite series; uytenbogaardtite; pearceite-polybasite solid solution; Kamchatka





## 1. Introduction

Volcanic belts on the Kamchatka Peninsula (Paleocene-Eocene Koryak-Western, Oligocene-Miocene Central-Kamchatka and Pliocene to recent East-Kamchatka) are well-known for being hosts of numerous precious metal deposits, including epithermal gold ores. Within the territory of the Kamchatka Peninsula, which comprises volcanogenic belts of different ages, up to 400 of both deposits and occurrences of gold are known [1]. All of them, including Rodnikovoe, belong to the LS type, with the exception of the Maletoyvayam deposit, located in the north of the Central Kamchatka volcanogenic belt. The Rodnikovoe deposit is located in the active hydrothermal zone to the north of the Mutnovsko-Asachinsk geothermal system in the area of the Vilyuchinsky hot springs, east of the Kamchatka volcanogenic belt (Figure 1) [1–6]. The Rodnikovoe deposit belongs to the low-sulfidation (LS) type [3] with a mineralization age (K-Ar) of 0.9–1.1 Ma. This deposit is comprehensively described in [6], where six main stages and several substages were stablished, including the physicochemical conditions of their formation. Furthermore, quartz, adularia, calcite, α-cristobalite, chlorite, illite and kaolinite are reported as the main gangue minerals.

Epithermal deposits not only represent an essential source of gold and silver but also of a few accessory elements, particularly, Se. Selenium is deemed a critical element for the high-tech industry and has seen an increased demand over the last few years. In this regard, comprehensive mineralogical and geochemical studies are fundamental to unveil

the parameters of ore-forming fluids that characterize and constrain endogenic ore deposits. The mode of occurrence, forming conditions, concentration parameters of noble metals and selenium, as well as their distribution in ores have numerous practical implications in prospecting and assessing new promising areas of epithermal Au-Ag deposits. Additionally, microthermometric studies, enhanced by gas chromatography-mass spectrometry, enable the evaluation of the role of hydrocarbons in the concentration of noble metals and selenium in ore-forming solutions, thus providing additional genetic aspects, such as the physicochemical (e.g., redox) conditions involved in ore formation. This work aims to meticulously study the mineralogical and geochemical features of the two main productive ore associations corresponding to different stages of ore formation. These features are undeniably key aspects in understanding gold metallogeny as a whole, and particularly the geochemical evolution of ore-forming systems. Moreover, new data, obtained in this study, may be useful for creating technological schemes in ore processing [1].

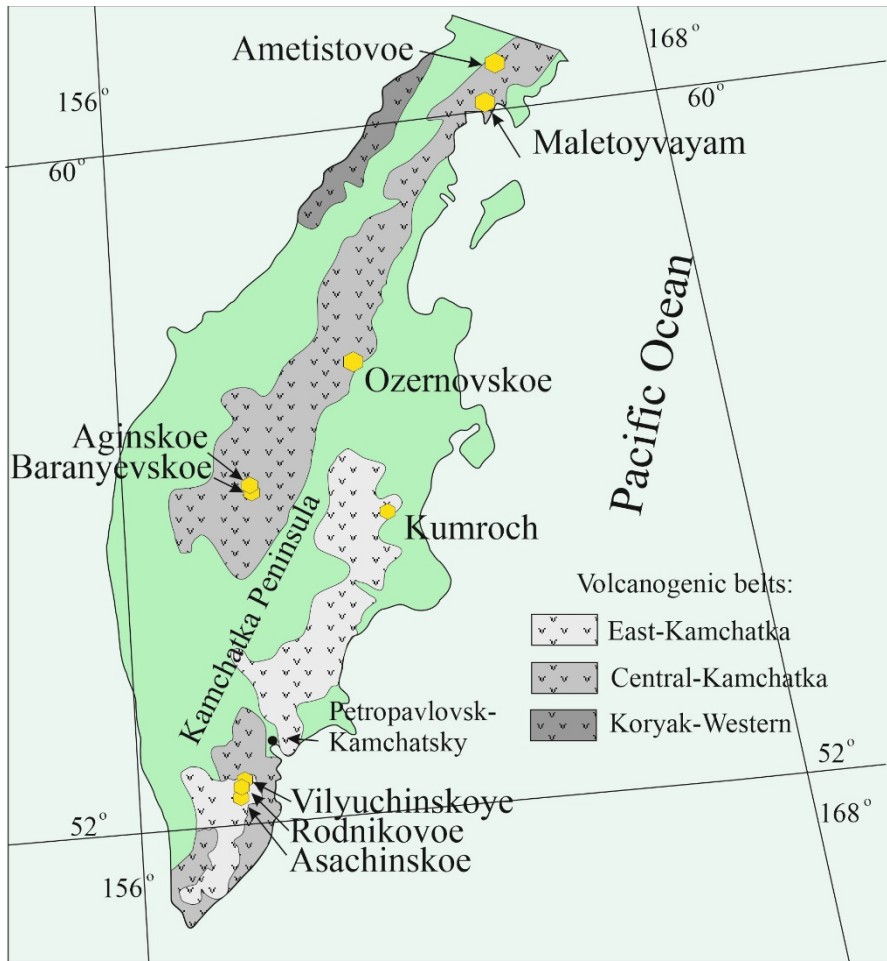

**Figure 1.** Map of volcanogenic belts showing the site of major gold deposits in the Kamchatka Peninsula [7]. East-Kamchatka: Quaternary basalts; Central-Kamchatka: Neogene andesites; Koryak-Western: Oligocene-Lower Miocene volcanogenic sediment [8].

## 2. Geological Background of the Rodnikovoe Deposit

The Mutnovsko-Asachinskaya geothermal zone is located 50–80 km south of Petropavlovsk-Kamchatsky. The volcanic rocks in the area were formed in three stages of volcanism: Oligocene to Miocene (andesite), Late Miocene to Pliocene (basalt, andesite and rhyolite) and Quaternary (basalt and andesite). Magmatic bodies related to the volcanism correspond to plutons and dikes of gabbro, diorite and andesite of Miocene to Pliocene

age [4,5]. Andesite is covered by ignimbrites of the Gorely volcano and basalt and andesitic lavas of the Vilyuchinsky volcano [3].

The ore field occupies an area of about 15 km$^2$, with two main industrial zones: Rodnikovoe and Vilyuchinskoe. The Rodnikovoe deposit is mainly confined to intrusive bodies of diorite and gabbro-diorite (Figure 2a), which may possibly represent a subvolcanic system of the Miocene to Pliocene ages [9]. Host rocks are composed of epidote-chlorite-actinolite-carbonate associations in propylites and argillites, whereas quartz, adularia, sericite and kaolinite are typical minerals of metasomatites [10]. The deposit itself is comprised by Au-Ag-bearing quartz-adularia and quartz-carbonate veins. The largest gold lode No. 44 has a length of 1.4 km, a thickness of 1–25 m and almost subvertical inclination. The average content of gold and silver is 10 ppm and 84 ppm, respectively, according to [11].

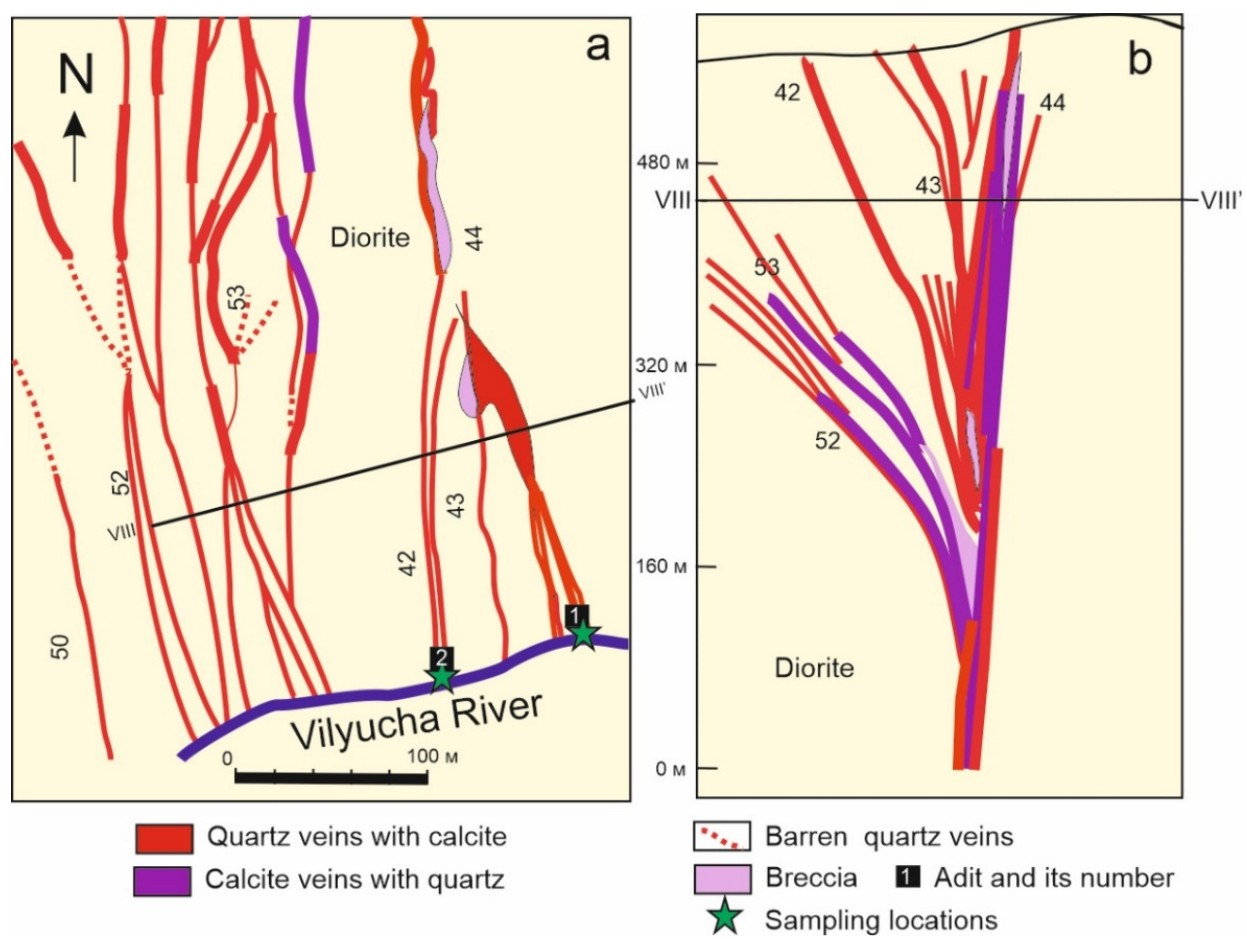

**Figure 2.** Geological scheme of the Rodnikovoe deposit (**a**) and cross section VIII-VIII′ (**b**) modified after [3,6,10]. The numbers correspond to the numbers of mineralized veins; the vertical scale indicates the height above mean sea level in meters.

The ore body is exposed on the northern bank of the Vilyucha River, which comprises a series of quartz-carbonate veins restricted to a submeridional fault. Samples were collected from mine dumps at adits 1 and 2 (Figure 2a), in which the main ore lodes have been exposed. There, the dip direction of major veins is almost vertical (80–90° W), whereas shorter veins located to the west of the major ones are inclined at 45–80° E (Figure 2b).

The hand specimen photographs of ore samples are represented by crustified-breccia textures (Figure 3). Fragments of a colomorphic and banded texture are visible in the sample (Figure 3a). Pyrite phenocrysts are found among the visible sulfides in the samples.

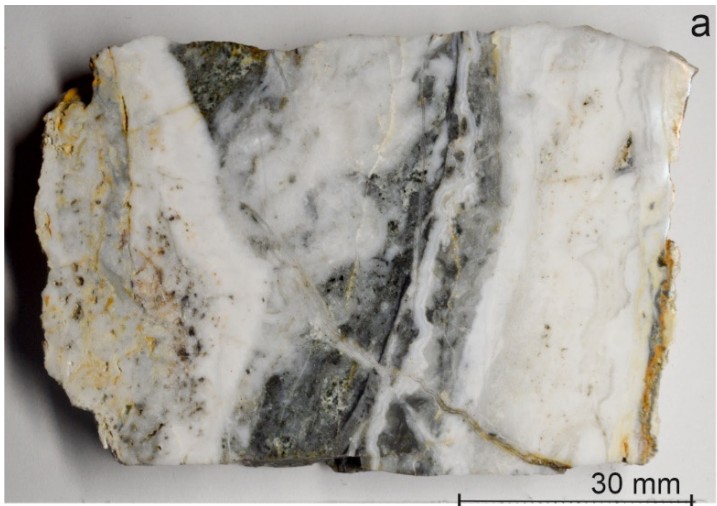
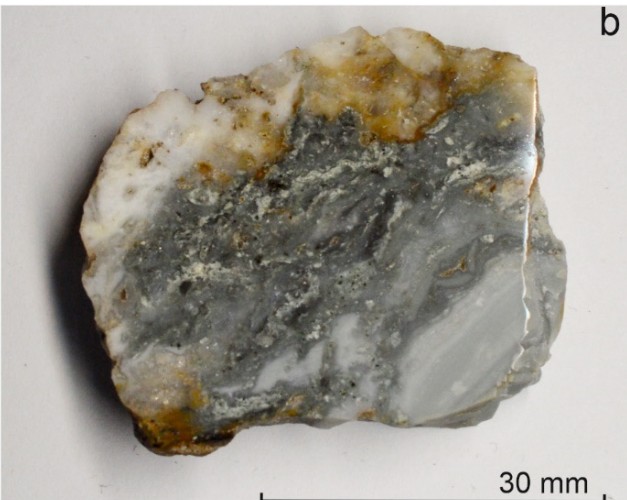

**Figure 3.** Sample of sericite-adularia-quartz composition with crustified-breccia texture containing disseminated sulfide: sample from adit 1 (**a**), sample from adit 2 (**b**).

## 3. Methods

The samples were investigated with an "Olympus" petrographic microscope during the first stage. Then, polished sections with grains from heavy concentrate were prepared for spectroscopic studies. The chemical composition was analyzed in a MIRA 3 LMU SEM (TESCAN Ltd., Brno, Czechia) equipped with an INCA Energy 450+ microprobe system, including an XMax-80 high-sensitive Si drift detector and a WDS INCA Wave 500 (Oxford Instruments Ltd., Abingdon, UK) (Analytical Center for Multi-Elemental and Isotope Research, Siberian Branch, Russian Academy of Sciences, Novosibirsk). The analyses were carried out at an accelerating voltage of 20 kV, a beam current of ~1.5 nA and a live time of 15–20 s. The standards included pure gold, $CuFeS_2$, PbSe and $Ag_2Te$ for Au, S, Se and Te, respectively (analysts N. Karmanov, M. Khlestov).

The amount of each element (except for Au) was determined on an inductively coupled plasma mass spectrometer (ICP-MS) at the JSC SGS testing laboratory of Vostok Limited, Chita branch. Au concentrations were obtained via inductively coupled plasma atomic emission spectroscopy (ICP-AES) in the same laboratory.

In order to assess the temperature, pressure and composition of ore-forming fluids, various methods were applied. Vapor-liquid and vapor-rich inclusions in quartz from quartz veins were analyzed using microthermometry—Raman spectroscopy in a Linkam TH-MSG-600 microthermal chamber with a measurement range from −196 to +600 °C according to the procedure described in [12]. The bulk composition of the gas component of the fluids was determined on a gas chromatography-mass spectrometer Focus GC/DSQ II MS (Thermo Scientific, Waltham, MA, USA). The GC-MS technique used is described in [13].

## 4. Results

### 4.1. Geochemical Features of Ores in the Rodnikovoe Deposit

Noble metals and associated chalcophile elements are shown in Table 1. Meanwhile the concentration of Ag prevails over Au: Au/Ag ratios vary from 0.08 to 0.18, which distinguishes the deposit as the gold-silver type. Some samples of the Rodnikovoe deposit are characterized by a small As anomaly (120 ppm), most likely due to As-bearing sulfosalts (pearceite). Tellurium, on the contrary, shows very low concentrations (0.05 ppm). Telluric mineralization, typical of many other LS epithermal deposits in the Chukotka region [14], is absent in the studied deposit (Table 2). The concentrations of other elements (Au, Ag, Cu, Bi, Sb, Sn, Se) are to a great extent comparable with mineralizations of the Chukotka district in the Arctic zone: Kupol, Dvoynoye, Sentyabrskoe, and Moroshka [15]. On the

other hand, the activity of Se is also significant (up to 25 ppm), owing to the presence of Se-bearing minerals (Table 2).

**Table 1.** Whole-rock concentration of ore and chalcophile elements of 3 samples from the Rodnikovoe deposit, ppm.

| Sample | Au | Ag | Au/Ag | Cu | As | Bi | Pb | Sb | Se | Sn | Zn | Te |
|---|---|---|---|---|---|---|---|---|---|---|---|---|
| Rodn-1 | 5.6 | 71.5 | 0.08 | 37.6 | 120 | 7.0 | 7.9 | 55.3 | 8.00 | 0.6 | 29 | 0.13 |
| Rodn-2 | 11 | 135 | 0.08 | 71.3 | 25.0 | 0.04 | 2.1 | 42.2 | 25.2 | 0.3 | 9.0 | 0.05 |
| Rodn-4 | 21 | 115 | 0.18 | 37.2 | 23.0 | 0.05 | 3.6 | 41.3 | 10.14 | 0.4 | 12.0 | 0.09 |

**Table 2.** Mineral assemblages of the gold-silver association in the Rodnikovoe deposit.

| Assemblage | Mineral Composition and Geochemical Speciation in Mineral Assemblages | |
|---|---|---|
| Ag-Au-aguilarite-acanthite | $(Ag,Au)$ <br> $Ag_2S$, $Ag_2(S,Se)$ <br> $Ag_4SeS$ <br> $Ag_3CuS_2$ <br> $Ag_9CuS_4(Ag,Au,Cu)_6(As,Sb)_2S_7$ <br> $Ag_6(Cu,Ag,Zn)_6(Sb,As)_4S_{13}$ <br> $Ag_6Cu_4(Zn,Fe)_2(Sb,As)_4S_{13}$ | Ag-Au alloys (Ag > Au) <br> Acanthite <br> Aguilarite <br> Jalpaite <br> Pearceite <br> Argentotetrahedrite-(Zn) <br> Argentotetrahedrite-(Zn,Fe) |
| Au-Ag-uytenbogaardtite-acanthite | $(Ag,Au)$ <br> $(Au,Ag)$ <br> $Ag_3AuS_2$ <br> $Ag_2Se$ <br> $Ag_3AsS_3$ <br> $Ag_9CuS_4(Ag,Au,Cu)_6(Sb,As)_2S_7$ <br> $Ag_5Cu_2S_5$ <br> $Ag_8GeS_6$ | Ag-Au alloys (Ag > Au) <br> Ag-Au alloys (Au > Ag) <br> Uytenbogaardtite <br> Naumannite <br> Proustite/Xanthoconite <br> Polybasite <br> Mckinstryite $(Ag_5Cu_3S_4)$? <br> Argyrodite |

### 4.2. Features of Mineral Associations

Sulfide minerals in the quartz-carbonate-adularia veins of the Rodnikovoe deposit are represented mainly by chalcopyrite, pyrite and galena, whereas either wurtzite, sphalerite or both, occur occasionally. Silver-bearing minerals, along with low-grade gold (20–60 μm) and native silver are the principal ores of the mineral association. Ag-bearing minerals appear intergrown with both pyrite and chalcopyrite, although more often with the latter.

In terms of mineralogical and geochemical features, the ore association of the Rodnikovoe deposit overall may be characterized as silver-chalcopyrite-quartz type, that according to its microparagenesis and representative minerals, is divided into two alternating assemblages (sub-stages): (1) silver-aguilarite-acanthite and (2) gold-uytenbogaardtite-acanthite assemblages (Table 2). Se-bearing acanthite is the most widespread mineral of the first assemblage, which also comprises aguilarite replacing Au-Ag alloys and jalpaite which replaces acanthite. The second assemblage is characterized by the presence of naumannite and argyrodite. Inclusions of pearceite-polybasite solid solutions are typical in both of the assemblages: with a recurrent prevalence of As varieties in the first series, and Sb varieties—in the second. The ore associations from both adits are basically the same, with only a few differences, such as the occurrence of jalpaite, which is more common in the early association from adit 2. Furthermore, both mineral associations are dominated by primarily large (50–70 μm) grains of Au-Ag alloys, with composition varying from 43 to 51 at.% Au (Figure 4). Secondary Au-Ag alloys develop along cracks and grain edges. The secondary alloys tend to be more Ag-rich in the first assemblage with their composition changing to native silver (2.3 at.% Au), whereas the fineness of secondary alloys, belonging to the second assemblage, increases up to 89 at.% Au (Table 3).

Typomorphic features recognized via photomicrograph analysis may be illustrated in the aguilarite-acanthite assemblage: (1) veinlets of silver in Au-Ag alloys (Figure 5a,b,d);

(2) grains of Au-Ag alloys are replaced by acanthite or aguilarite (borders around grains) (Figure 5a–d), the composition of which is determined by the intermediate solid solutions between these two minerals; (3) acanthite is replaced by jalpaite (spotted textures) (Figure 5g,h) with the addition of copper from hydrothermal solutions and the release of silver in its pure form, the inclusions of which are also present as shown in the reaction: $2Ag_2(S,Se) + Cu = Ag_3Cu(S,Se)_2 + Ag^o$; and (4) with pearceite prevailing over polybasite (Figure 5c,e,f), and Zn,Ag-tetrahedrite present as well (Figure 5i). The following features are distinctive of the gold-uytenbogaardtite-acanthite assemblage: (1) thinnest rims of high-grade gold in grains of Ag-Au alloys (Figure 6a,c–e); (2) repeated replacement of Au-Ag alloys by uytenbogaardtite (Figure 5a–d) and acanthite (Figure 6c,d,f); (3) polybasite prevailing over pearceite (Figure 6b,e); and (4) occurrence of naumannite and argyrodite $(Ag_8GeS_6)$.

**Table 3.** Composition of some representative Au-Ag alloys of the Rodnikovoe deposit.

| No. | Au | Ag | Total | Au | Ag | Total | No. | Au | Ag | Total | Au | Ag | Total |
|---|---|---|---|---|---|---|---|---|---|---|---|---|---|
| | **wt.%** | | | **at.%** | | | | **wt.%** | | | **at.%** | | |
| 1 | 3.07 | 97.3 | 100.4 | 1.70 | 98.30 | 100 | 40 | 61.89 | 38.33 | 100.22 | 46.93 | 53.07 | 100 |
| 2 | 9.48 | 89.5 | 98.96 | 5.48 | 94.52 | 100 | 41 | 62.63 | 38.15 | 100.78 | 47.34 | 52.66 | 100 |
| 3 | 19.61 | 81.2 | 100.8 | 11.68 | 88.32 | 100 | 42 | 61.49 | 37.23 | 98.72 | 47.49 | 52.51 | 100 |
| 4 | 19.36 | 79.6 | 98.92 | 11.76 | 88.24 | 100 | 43 | 62.19 | 37.15 | 99.34 | 47.83 | 52.17 | 100 |
| 5 | 24.89 | 74.6 | 99.5 | 15.45 | 84.55 | 100 | 44 | 63.01 | 37.55 | 100.56 | 47.89 | 52.11 | 100 |
| 6 | 26.92 | 72.2 | 99.16 | 16.95 | 83.05 | 100 | 45 | 62.52 | 37.19 | 99.71 | 47.93 | 52.07 | 100 |
| 7 | 30.97 | 69.9 | 100.9 | 19.53 | 80.47 | 100 | 46 | 62.25 | 36.9 | 99.15 | 48.02 | 51.98 | 100 |
| 8 | 31.54 | 68.7 | 100.2 | 20.10 | 79.90 | 100 | 47 | 62.97 | 37.31 | 100.28 | 48.03 | 51.97 | 100 |
| 9 | 31.26 | 67.2 | 98.5 | 20.29 | 79.71 | 100 | 48 | 63.44 | 37.54 | 100.98 | 48.07 | 51.93 | 100 |
| 10 | 35.00 | 65.6 | 100.6 | 22.61 | 77.39 | 100 | 49 | 63.34 | 37.43 | 100.77 | 48.10 | 51.90 | 100 |
| 11 | 38.79 | 60.3 | 99.05 | 26.06 | 73.94 | 100 | 50 | 63.09 | 37.28 | 100.37 | 48.10 | 51.90 | 100 |
| 12 | 39.31 | 60.6 | 99.94 | 26.20 | 73.80 | 100 | 51 | 62.50 | 36.4 | 98.9 | 48.46 | 51.54 | 100 |
| 13 | 43.16 | 57.1 | 100.2 | 29.28 | 70.72 | 100 | 52 | 62.91 | 36.6 | 99.51 | 48.49 | 51.51 | 100 |
| 14 | 44.87 | 54.6 | 99.44 | 31.05 | 68.95 | 100 | 53 | 62.47 | 36.32 | 98.79 | 48.51 | 51.49 | 100 |
| 15 | 44.81 | 54.4 | 99.18 | 31.10 | 68.90 | 100 | 54 | 63.34 | 36.74 | 100.08 | 48.56 | 51.44 | 100 |
| 16 | 45.82 | 54.1 | 99.9 | 31.69 | 68.31 | 100 | 55 | 63.8 | 36.59 | 100.39 | 48.85 | 51.15 | 100 |
| 17 | 49.56 | 49.9 | 99.43 | 35.24 | 64.76 | 100 | 56 | 63.01 | 36.12 | 99.13 | 48.86 | 51.14 | 100 |
| 18 | 49.97 | 48.2 | 98.15 | 36.22 | 63.78 | 100 | 57 | 62.50 | 35.67 | 98.17 | 48.97 | 51.03 | 100 |
| 19 | 51.58 | 47.8 | 99.37 | 37.15 | 62.85 | 100 | 58 | 62.54 | 35.62 | 98.16 | 49.02 | 50.98 | 100 |
| 20 | 54.05 | 46.2 | 100.3 | 39.04 | 60.96 | 100 | 59 | 63.65 | 36.07 | 99.72 | 49.15 | 50.85 | 100 |
| 21 | 57.43 | 43.5 | 100.9 | 41.96 | 58.04 | 100 | 60 | 63.76 | 36.13 | 99.89 | 49.15 | 50.85 | 100 |
| 22 | 58.03 | 42.3 | 100.4 | 42.88 | 57.12 | 100 | 61 | 64.09 | 36.04 | 100.13 | 49.34 | 50.66 | 100 |
| 23 | 58.26 | 42 | 100.3 | 43.15 | 56.85 | 100 | 62 | 64.17 | 35.87 | 100.04 | 49.49 | 50.51 | 100 |
| 24 | 57.30 | 41.2 | 98.49 | 43.24 | 56.76 | 100 | 63 | 63.35 | 35.31 | 98.66 | 49.56 | 50.44 | 100 |
| 25 | 58.21 | 41.8 | 99.96 | 43.30 | 56.70 | 100 | 64 | 64.54 | 35.6 | 100.14 | 49.82 | 50.18 | 100 |
| 26 | 57.45 | 40.6 | 98.06 | 43.65 | 56.35 | 100 | 65 | 64.22 | 35.09 | 99.31 | 50.06 | 49.94 | 100 |
| 27 | 57.94 | 40.7 | 98.59 | 43.84 | 56.16 | 100 | 66 | 64.77 | 34.23 | 99.00 | 50.89 | 49.11 | 100 |
| 28 | 59.03 | 40.6 | 99.58 | 44.36 | 55.64 | 100 | 67 | 65.40 | 34.34 | 99.74 | 51.05 | 48.95 | 100 |
| 29 | 58.74 | 40.3 | 99.02 | 44.40 | 55.60 | 100 | 68 | 65.96 | 34.63 | 100.59 | 51.05 | 48.95 | 100 |
| 30 | 60.14 | 40.7 | 100.9 | 44.72 | 55.28 | 100 | 69 | 66.63 | 33.36 | 99.99 | 52.24 | 47.76 | 100 |
| 31 | 59.42 | 39.8 | 99.23 | 44.98 | 55.02 | 100 | 70 | 67.67 | 32.86 | 100.53 | 53.00 | 47.00 | 100 |
| 32 | 58.87 | 39.3 | 98.12 | 45.10 | 54.90 | 100 | 71 | 67.46 | 32.27 | 99.73 | 53.38 | 46.62 | 100 |
| 33 | 59.04 | 39.1 | 98.16 | 45.25 | 54.75 | 100 | 72 | 67.33 | 32.17 | 99.5 | 53.41 | 46.59 | 100 |
| 34 | 59.96 | 39.2 | 99.19 | 45.56 | 54.44 | 100 | 73 | 75.59 | 25.00 | 100.59 | 62.35 | 37.65 | 100 |
| 35 | 60.78 | 39.6 | 100.4 | 45.69 | 54.31 | 100 | 74 | 74.58 | 24.44 | 99.02 | 62.56 | 37.44 | 100 |
| 36 | 60.20 | 39 | 99.16 | 45.83 | 54.17 | 100 | 75 | 77.73 | 21.85 | 99.58 | 66.08 | 33.92 | 100 |
| 37 | 61.74 | 38.6 | 100.3 | 46.70 | 53.30 | 100 | 76 | 77.33 | 21.09 | 98.42 | 66.76 | 33.24 | 100 |
| 38 | 61.49 | 38.3 | 99.83 | 46.76 | 53.24 | 100 | 77 | 81.38 | 19.2 | 100.58 | 69.89 | 30.11 | 100 |
| 39 | 60.97 | 37.8 | 98.77 | 46.90 | 53.10 | 100 | 78 | 92.73 | 6.46 | 99.19 | 88.71 | 11.29 | 100 |

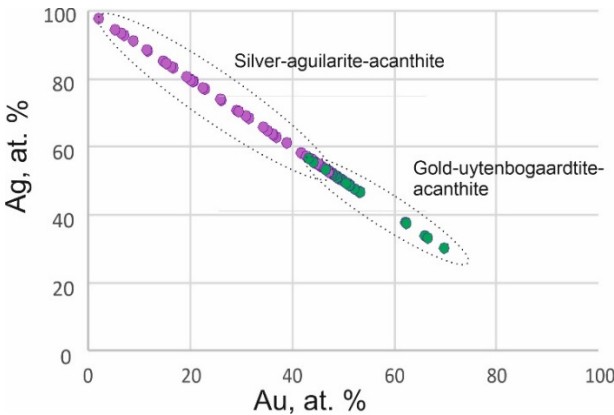

**Figure 4.** Variations in the compositions of Au-Ag alloys from the Rodnikovoe deposit, showing the main two established assemblages (sub-stages).

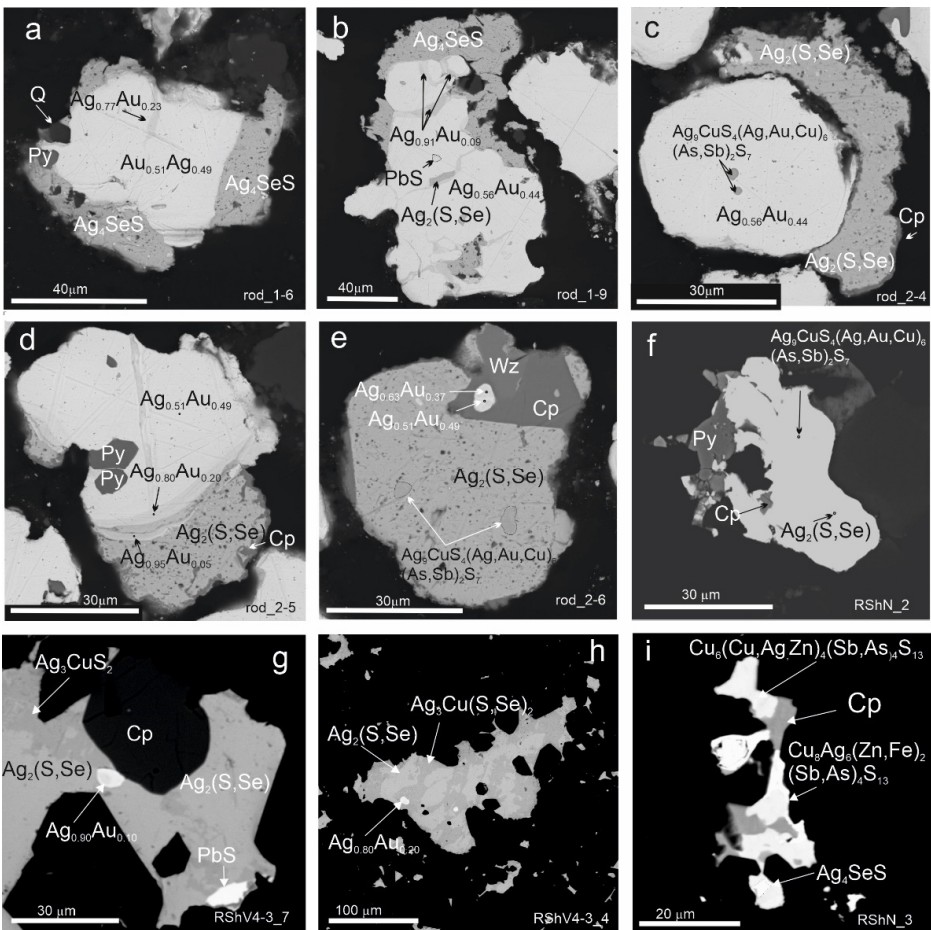

**Figure 5.** SEM-BSE images of ore concentrates from the Rodnikovoe deposit. The silver-aguilarite-acanthite assemblage: (**a–d**) Electrum (Au-Ag) grains bordered by aguilarite ($Ag_4SeS$) or Se-acanthite [$Ag_2(S,Se)$], including veins and borders of Ag-rich alloy (**a,b,d**) and inclusions of galena (PbS) (**b**), pyrite (Py) (**a,d**), and pearceite $Ag_9CuS_4(Ag,Au,Cu)_6(As,Sb)_2S_7$ (**c**); (**e**) acanthite grain with inclusions of pearceite, intergrown with chalcopyrite (Cp) and wurtzite (Wz); (**f**) intergrowth of acanthite-pyrite and inclusions of pearceite; (**g,h**) replacement texture of acanthite by jalpaite $Ag_3CuS_2$ with inclusions of native silver and galena; (**i**) Ag,Zn-tetrahedrite $Cu_6(Cu,Ag,Zn)_4(Sb,As)_4S_{13}$ and $Cu_8Ag_6(Zn,Fe)_2(Sb,As)_4S_{13}$ intergrown with chalcopyrite and inclusion of aguilarite.

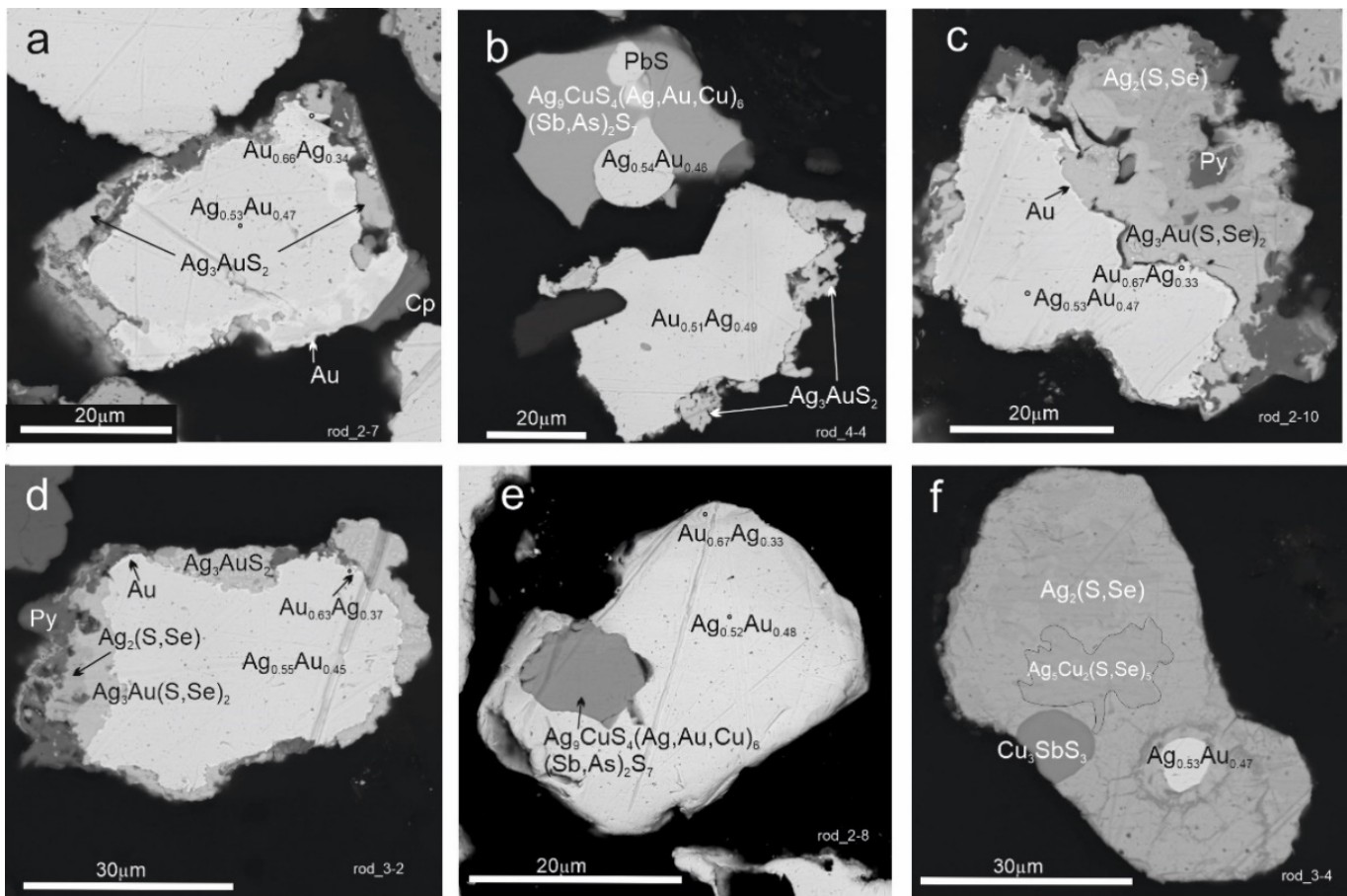

**Figure 6.** SEM-BSE images of ore concentrates from the Rodnikovoe deposit. The gold-uytenbogaardtite-acanthite assemblage: (**a**,**c**,**d**) Grains of Ag-Au alloy featuring thin rims of higher-grade gold, replaced by an aggregate of uytenbogaardtite $Ag_3Au(S,Se)_2$ and acanthite $Ag_2(S,Se)$; (**b**) drop-shaped inclusions of galena (PbS) and Ag-Au alloy in polybasite $[Ag_9CuS_4Ag,Au,Cu)_6(Sb,As)_2S_7]$, combined with replacement of Au-Ag by uytenbogaardtite; (**e**) Ag-Au alloy with inclusion of pearceite; (**f**) acanthite grain partially replaced by $Ag_5Cu_2S_5$ close to mckinstryite with drop-shaped inclusions of skinnerite $Cu_3SbS_3$ and relicts of Ag-Au alloy.

### 4.3. Composition of Ag-Au Minerals in the Rodnikovoe Deposit

Acanthite displays a complete solid solutions series from acanthite $Ag_2S$ to aguilarite $Ag_4SeS$ (up to 14.89 wt.% Se) according to the S − Ag+(Au,Cu) − Se ternary plot. The compositions vary from $Ag_{0.97}(S_{0.96}Se_{0.07})_{1.03}$ to $Ag_{2.01}(Se_{0.53}S_{0.46})_{0.99}$. Meanwhile, compositions between aguilarite and naumannite $Ag_2Se$ were not found. Naumannite contains minor sulfur (0.71 wt.%) (Table 4) and its compositions show a regular deviation from the $Ag_2S \rightarrow Ag_2Se$ trend: with an increase in selenium, the concentration of silver rises, and a new trend is developed: $Ag_{1.91}(S_{0.95}Se_{0.08})_{1.06} \rightarrow Ag_{2.08}(Se_{0.86}S_{0.06})_{0.92}$ or $Ag_{2-x}S_{1+x} \rightarrow Ag_{2+x}Se_{1-x}$ (Figure 7a).

Uytenbogaardtite, $Ag_3AuS_2$, rarely coincides with stoichiometric compositions (Table 5), forming a limited trend in the Ag − Au − (S+Se) system with significant variations in the concentrations of all elements. An approximate trend lies on the line connecting native silver with a hypothetical composition $AuS_2$ (Figure 7b).

**Table 4.** Compositions and empirical formulae of acanthite, aguilarite and naumannite, wt.%.

| No. | Sample | Au | Ag | Se | S | Total | Formula |
|---|---|---|---|---|---|---|---|
| 1 | Rod-4_3 | | 83.68 | 2.15 | 12.05 | 97.88 | $Ag_{1.97}(S_{0.96}Se_{0.07})_{1.03}$ |
| 2 | Rod-4_3 | | 83.87 | 2.44 | 12.52 | 98.83 | $Ag_{1.95}(S_{0.98}Se_{0.08})_{1.06}$ |
| 3 | Rod-3_1 | | 83.21 | 4.38 | 12.37 | 99.96 | $Ag_{1.91}(S_{0.95}Se_{0.14})_{1.09}$ |
| 4 | Rod-2_10 | 2.32 | 80.82 | 5.25 | 10.36 | 98.75 | $(Ag_{1.95}Au_{0.03})_{1.98}(S_{0.84}Se_{0.17})_{1.01}$ |
| 5 | Rod-2_7 | 1.96 | 79.96 | 5.68 | 10.36 | 97.96 | $(Ag_{1.94}Au_{0.03})_{1.97}(S_{0.85}Se_{0.19})_{1.04}$ |
| 6 | Rod-2_5 | | 82.3 | 6.25 | 10.43 | 98.98 | $Ag_{1.96}(S_{0.84}Se_{0.20})_{1.04}$ |
| 7 | Rod-3_3 | | 82.92 | 6.31 | 10.34 | 99.57 | $Ag_{1.97}(S_{0.83}Se_{0.20})_{1.03}$ |
| 8 | Rod-2_1 | | 81.61 | 6.56 | 10.55 | 98.72 | $Ag_{1.94}(S_{0.84}Se_{0.21})_{1.05}$ |
| 9 | Rod-3_1 | | 82.65 | 6.92 | 10.53 | 100.1 | $Ag_{1.94}(S_{0.83}Se_{0.22})_{1.05}$ |
| 10 | Rod-2_1 | | 82.23 | 7.21 | 10.26 | 99.7 | $Ag_{1.95}(S_{0.82}Se_{0.23})_{1.05}$ |
| 11 | Rod-2_4 | | 80.94 | 7.31 | 9.36 | 97.61 | $Ag_{1.98}(S_{0.77}Se_{0.24})_{1.01}$ |
| 12 | Rod-2_6 | | 79.69 | 8.26 | 9.06 | 97.96 * | $Ag_{1.94}(S_{0.74}Se_{0.28})_{1.02}$ |
| 13 | Rod-3_2 | | 79.35 | 10.01 | 8.65 | 98.01 | $Ag_{1.95}(S_{0.71}Se_{0.34})_{1.05}$ |
| 14 | Rod-1_9 | 1.91 | 79.23 | 9.91 | 7.2 | 98.25 | $Ag_{4.03}Se_{0.69}S_{1.23}$ |
| 15 | Rod-1_1a | | 78.51 | 11.35 | 6.8 | 96.66 | $Ag_{4.01}Se_{0.79}S_{1.17}$ |
| 16 | Rod-1_5 | | 78.74 | 11.72 | 7.23 | 97.69 | $Ag_{3.97}Se_{0.81}S_{1.23}$ |
| 17 | Rod-1_1b | | 79.71 | 11.92 | 7.11 | 98.74 | $Ag_{3.99}Se_{0.81}S_{1.20}$ |
| 18 | Rod-1_1c | | 79.29 | 12.33 | 6.83 | 98.45 | $Ag_{3.99}Se_{0.85}S_{1.16}$ |
| 19 | Rod-1_1c | | 79.34 | 12.88 | 7.06 | 99.28 | $Ag_{3.94}Se_{0.87}S_{1.18}$ |
| 20 | Rod-1_7 | | 78.04 | 14.89 | 5.28 | 98.21 | $Ag_{4.03}Se_{1.05}S_{0.92}$ |
| 21 | Rod-1_8 | | 76.56 | 23.08 | 0.71 | 100.4 | $Ag_{2.08}(Se_{0.86}S_{0.06})_{0.92}$ |
| 22 | Rod-1_1c | | 73.44 | 24.59 | 0.41 | 98.44 | $Ag_{2.03}(Se_{0.93}S_{0.04})_{0.97}$ |

1–13—acanthite $Ag_2S$; 14–20—aguilarite $Ag_4SeS$; 21,22—naumannite. *—the total includes 0.95 wt.% Cu.

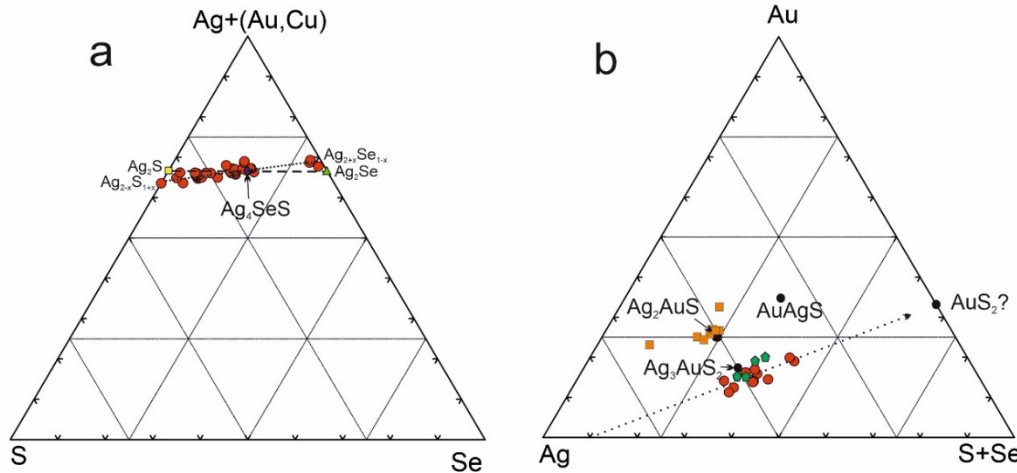

**Figure 7.** Composition of Ag-Au minerals: (**a**) Acanthite–aguilarite $Ag_2S$-$Ag_4SeS$ solid solution and a few naumannite $Ag_2Se$ samples; (**b**) ternary graph of the Ag – Au – (S+Se) system: $Ag_3AuS_2$—uytenbogaardtite, AuAgS—petrovskaite, $Ag_2AuS$—unknown phase (black dots). Green hexagons—uytenbogaardtite from the Valunistoye deposit [16]; orange squares—"Uytenbogaardtite" from the Julieta deposit [17].

The minerals of the pearceite-polybasite solid solution have a complex chemical formula. Their composition is presented in Table 6. A simplified formula is shown in Figure 8a. Minerals of pearceite-polybasite group according to [18] refer to pearceite, antimonpearceite, polybasite and arsenpolybasite. This phase contains Au in some analyses up to 12.2 wt.% (Table 6) and shows a slight variation between (Ag, Cu, Au) and (S + Se) (Figure 8a). The total amount of As and Sb vary from 1.66 to 1.91 apfu; both Sb-rich varieties (polybasite) and As-rich varieties (pearceite) are shown in (Figure 8b).

**Table 5.** Composition of minerals of the Au–Ag(Cu)–S(Se) system close to uytenbogaardtite composition see Figure 6b, wt.%.

| No. | Sample | Cu | Au | Ag | Se | S | Total | Formula |
|---|---|---|---|---|---|---|---|---|
| 1 | Rod-2_7 | 2.10 | 23.58 | 59.39 | 2.59 | 10.46 | 98.12 | $(Ag_{3.11}Cu_{0.19})_{3.30}Au_{0.68}(S_{1.84}Se_{0.19})_{2.03}$ |
| 2 | Rod-2_10 | 0.92 | 25.54 | 58.49 | 2.21 | 10.56 | 97.72 | $(Ag_{3.12}Cu_{0.08})_{3.20}Au_{0.75}(S_{1.89}Se_{0.16})_{2.05}$ |
| 3 | Rod-3_2 | 0.53 | 27.07 | 57.78 | 0.00 | 10.64 | 96.02 | $(Ag_{3.17}Cu_{0.05})_{3.22}Au_{0.81}S_{1.97}$ |
| 4 | Rod-3_2 | 1.01 | 28.57 | 53.30 | 4.02 | 11.03 | 97.93 | $(Ag_{2.82}Cu_{0.09})_{2.91}Au_{0.83}(S_{1.97}Se_{0.29})_{2.26}$ |
| 5 | Rod-2_10 | 0.84 | 29.84 | 54.74 | 3.55 | 11.58 | 100.55 | $(Ag_{2.82}Cu_{0.07})_{2.89}Au_{0.84}(S_{2.01}Se_{0.25})_{2.26}$ |
| 6 | Rod-3_2 | 0.67 | 28.32 | 58.66 | 0.00 | 10.22 | 97.87 | $(Ag_{3.21}Cu_{0.06})_{3.27}Au_{0.85}S_{1.88}$ |
| 7 | Rod-2_1 | 1.45 | 31.58 | 51.35 | 0.00 | 14.39 | 98.77 | $(Ag_{2.58}Cu0.12)_{2.70}Au_{0.87}S_{2.43}$ |
| 8 | Rod-3_5 | 0.41 | 33.73 | 53.51 | 0.00 | 12.96 | 100.61 | $(Ag_{2.76}Cu_{0.04})_{2.80}Au_{0.95}S_{2.25}$ |
| 9 | Rod-1_1c | 0.00 | 32.31 | 53.30 | 1.41 | 10.75 | 97.77 | $Ag_{2.93}Au_{0.97}(S_{1.99}Se_{0.11})_{2.10}$ |
| 10 | Rod-4_4 | 0.00 | 30.03 | 49.53 | 0.00 | 9.50 | 89.06 | $Ag_{3.03}Au_{1.01}S_{1.96}$ |
| 11 | Rod-3_2 | 0.53 | 33.93 | 49.55 | 0.00 | 12.14 | 96.15 | $(Ag_{2.71}Cu_{0.05})_{2.76}Au_{1.01}S_{2.23}$ |
| 12 | Rod-2_10 | 0.70 | 34.37 | 50.53 | 2.64 | 10.95 | 99.19 | $(Ag_{2.72}Cu_{0.06})_{2.78}Au_{1.01}(S_{1.99}Se_{0.19})_{2.18}$ |
| 13 | Rod-2_7 | 2.05 | 40.43 | 39.75 | 1.55 | 14.50 | 98.28 | $(Ag_{2.05}Cu_{0.18})_{2.23}Au_{1.14}(S_{2.52}Se_{0.11})_{2.63}$ |
| 14 | Rod-1_1c | 0.00 | 41.80 | 43.28 | 1.75 | 13.76 | 100.59 | $Ag_{2.26}Au_{1.20}(S_{2.42}Se_{0.12})_{2.54}$ |

**Table 6.** Composition of pearceite-polybasite solid solutions (wt.%), see Figure 7a,b.

| No. | Sample | Cu | Au | Ag | Sb | As | Se | S | Total | Formula |
|---|---|---|---|---|---|---|---|---|---|---|
| 1 | Rod-2_8 | 7.47 | 1.85 | 69.30 | 0.00 | 5.69 | 1.23 | 15.02 | 100.56 | $Ag_{9.00}Cu_{1.00}S_{4.00}(Ag_{5.01}Cu_{1.56}Au_{0.20})_{6.77}As_{1.66}(S_{6.22}Se_{0.34})_{6.56}$ |
| 2 | Rod-2_6 | 8.80 | | 65.55 | 2.12 | 5.41 | 2.76 | 15.28 | 99.92 | $Ag_{9.00}Cu_{1.00}S_{4.00}(Ag_{4.08}Cu_{1.98})_{6.06}(As_{1.55}Sb_{0.37})_{1.92}(S_{6.26}Se_{0.75})_{7.01}$ |
| 3 | Rod-2_4 | 8.95 | 4.19 | 60.56 | 2.31 | 4.87 | 1.72 | 14.46 | 97.06 | $Ag_{9.00}Cu_{1.00}S_{4.00}(Ag_{3.72}Cu_{2.19}Au_{0.48})_{6.39}(As_{1.47}Sb_{0.43})_{1.90}(S_{6.22}Se_{0.49})_{6.71}$ |
| 4 | Rod-2_4 | 7.68 | 12.20 | 55.77 | 2.97 | 3.76 | 2.01 | 12.60 | 96.99 | $Ag_{9.00}Cu_{1.00}S_{4.00}(Ag_{3.57}Cu_{1.94}Au_{1.51})_{7.02}(As_{1.22}Sb_{0.59})_{1.81}(S_{5.55}Se_{0.62})_{6.17}$ |
| 5 | Rod-4_3 | 2.54 | | 75.08 | 4.71 | 2.77 | 1.32 | 12.99 | 99.41 | $Ag_{9.00}Cu_{0.94}S_{4.00}Ag_{7.36}(As_{0.87}Sb_{0.91})_{1.78}(S_{5.53}Se_{0.39})_{5.92}$ |
| 6 | Rod-2_8 | 6.38 | 2.24 | 67.91 | 6.57 | 2.22 | 1.42 | 14.30 | 101.04 | $Ag_{9.00}Cu_{1.00}S_{4.00}(Ag_{5.16}Cu_{1.26}Au_{0.26})_{6.68}(Sb_{1.21}As_{0.67})_{1.88}(S_{6.40}Se_{0.40})_{6.44}$ |
| 7 | Rod-4_4 | 7.76 | | 67.80 | 6.54 | 1.70 | 1.65 | 13.78 | 99.23 | $Ag_{9.00}Cu_{1.00}S_{4.00}(Ag_{5.27}Cu_{1.77})_{7.04}(Sb_{1.22}As_{0.51})_{1.73}(S_{5.76}Se_{0.47})_{6.23}$ |
| 8 | | 7.65 | | 69.41 | 6.88 | 1.89 | 1.74 | 14.45 | 102.02 | $Ag_{9.00}Cu_{1.00}S_{4.00}(Ag_{5.15}Cu_{1.65})_{6.80}(Sb_{1.24}As_{0.55})_{1.79}(S_{5.91}Se_{0.48})_{6.39}$ |
| 9 | | 6.43 | | 69.64 | 6.79 | 1.41 | 1.33 | 12.25 | 97.85 | $Ag_{9.00}Cu_{1.00}S_{4.00}(Ag_{6.34}Cu_{1.40})_{7.74}(Sb_{1.33}As_{0.45})_{1.78}(S_{5.08}Se_{0.40})_{5.48}$ |
| 10 | Rod-3_6 | 1.38 | 5.45 | 69.57 | 7.06 | 1.66 | 4.95 | 12.21 | 102.28 | $Ag_{9.00}Cu_{0.52}S_{4.00}(Ag_{6.36}Au_{0.66})_{7.02}(Sb_{1.38}As_{0.53})_{1.91}(S_{5.07}Se_{1.49})_{6.56}$ |
| 11 | Rod-2_6 | 8.24 | | 65.03 | 7.62 | 1.40 | 3.00 | 14.49 | 99.78 | $Ag_{9.00}Cu_{1.88}S_{4.00}(Ag_{4.41}Cu_{1.88})_{6.29}(Sb_{1.39}As_{0.42})_{1.81}(S_{6.05}Se_{0.85})_{6.90}$ |

Analyses with high gold content possibly contaminated by micro-inclusions of native gold.

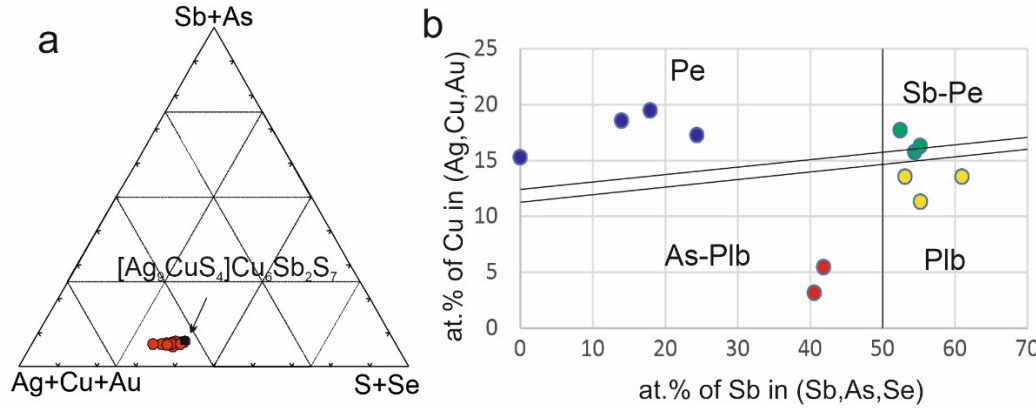

**Figure 8.** Compositions of the pearceite-polybasite solid solution: (**a**) Compositions obtained in this study (red circles) compared with the idealized empirical formula (black circle); (**b**) relationship between the at.% of Sb in (Sb, As, Se) and the at.% of Ag in (Ag, Cu, Cu), and minerals of the pearceite-polybasite group according nomenclature [18]: Pe—pearceite, Sb-Pe—antimonpearceite, Plb—polybasite, As-Plb—arsenpolybasite.

Selenium is ubiquitous in all the studied minerals but in variable amounts: its highest concentration is found in acanthite (Table 4), where it progressively replaces sulfur until it reaches the aguilarite composition. There is a clear negative correlation between S and Se within this series (Figure 9a). The analyzed compositions of uytenbogaardtite contain a minimum amount of selenium (up to 4.02 wt.%), substituting sulfur into the structure

of the mineral (Table 5), and a rather weak S-Se negative correlation (Figure 9b). Whereas, the Se concentrations in pearceite-polybasite minerals reach up to 4.95 wt.%, but no S-Se correlation is observed (Figure 9c).

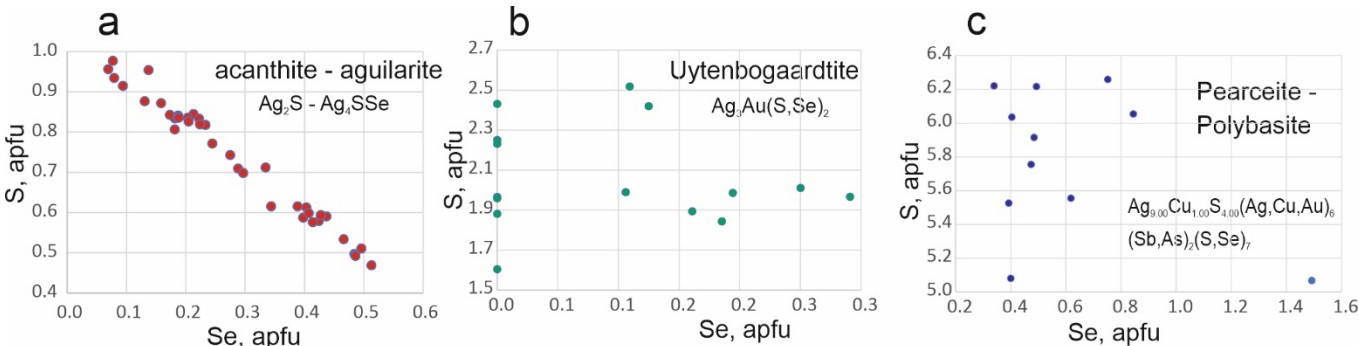

**Figure 9.** Sulfur versus selenium content (apfu) in the minerals acanthite-aguilarite (**a**), uytenbogaardtite (**b**), pearceite-polybasite (**c**).

*4.4. Composition and Physicochemical Parameters of Hydrothermal Fluids*

Microthermometric studies of gas-liquid and gas inclusions in quartz, from quartz veins, of the Rodnikovoe deposit indicated homogenization temperatures between 265 and 160 °C with a salinity range within 0.8–2.5 wt.% NaCl equiv., and pressure of 6–50 bar. The gas phase composition was established by gas chromatography-mass spectrometry (GC-MS): $H_2O$ (95 rel.%) and $CO_2$ (4 rel.%), and detected a wide range of organic compounds (OC) in the ore-forming fluids: 155 substances, about 1 rel.% (Figure 10a), including cyclic hydrocarbons, as well as sulfonated and nitrogenated compounds, represented by oxygen-free aliphatic (paraffins, olefins), cyclic hydrocarbons (cycloalkanes and cycloalkenes, arenes, polycyclic aromatic hydrocarbons) and oxygenated hydrocarbons (alcohols, ethers and esters, furans, aldehydes, ketones, carboxylic acids) (Figure 10b).

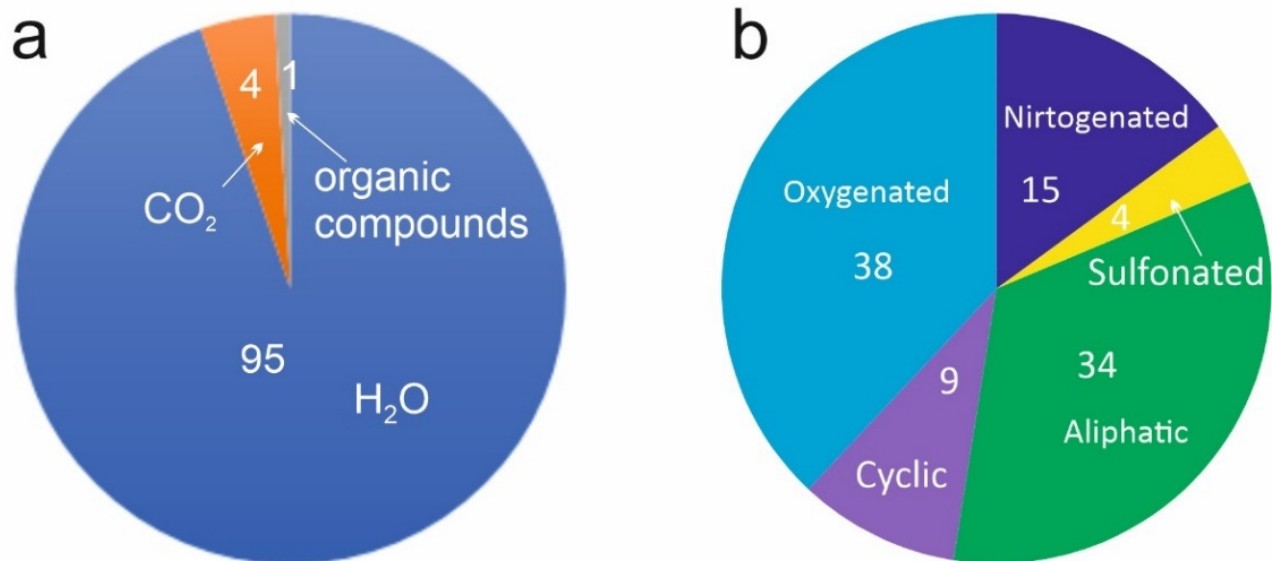

**Figure 10.** Composition of gas-liquid inclusions in quartz from the ore assemblage of the Rodnikovoe deposit; (**a**) gases and organic compounds (OC), relative %; (**b**) organic compounds only in relative %.

## 5. Discussion

*5.1. The Role of Organic Compounds in the Accumulation of Selenium*

The Rodnikovoe deposit is a classic example of epithermal deposits of the low sulfidation type. In terms of mineral associations, it is comparable with many other Au-Ag

deposits of the East Chukotka flank zone (Valunistoye, Zhilnoye, Corrida, Pepenveem) and the Central Chukotka area (Kupol, Kapel'ka, Promezhutochnoe) in the Chukotka volcanic belt [14]. Rodnikovoe also shares some features with the Julietta deposit in Northeast Russia [17], among others. These deposits are characterized by the presence of Se-acanthite—S-naumannite solid solution along with other Se-bearing minerals. Similarly, selenium plays a significant role at the Rodnikovoe deposit, where there is a complete solid solution series from acanthite to aguilarite (up to 12.88 wt.% Se), as well as S-naumannite. In addition, uytenbogaardtite contains Se up to 4.02 wt.%, and the minerals of the pearceite-polybasite solid solutions up to 4.95 wt.% Se.

According to [6], the ore-forming temperature range of the Rodnikovoe gold deposit is about 250–150 °C, which is to a great extent comparable with the homogenization temperatures (265–160 °C) obtained in this study. At temperatures around 200 °C, in solutions close to neutral or weakly alkaline, gold may be dissolved in the form of $Au(HS)_2^-$ complexes [19], whereas selenium could migrate as selenomethionine ($CH_3$–$SeCH_2$–$CH_2NH_2$–$COOH$) and selenocysteine ($HSeCH_2$–$CHNH_2$–$COOH$) [20]. These ligand organic complexes are transported by ascending hydrothermal flows [6], decompose, releasing selenium, which then participates in the formation of inorganic compounds (selenites, selenates, selenides), releasing hydrocarbons. The presence of organic compounds in fluid inclusions implies a role in the concentration and mobilization of ore elements, including selenium [21]. Selenium is a good ligand for silver, and probably other ore elements, since Se is usually correlated positively with noble metals and chalcogens. However, it is worth noting that the contribution of selenium to the collection of metals in ore-forming events is higher with decreasing temperature [22].

Both bacteria and algae are the primary sources of organic matter, that once modified, may accumulate selenium, enriching intermediate-depth reservoirs [21]. These organic complexes may eventually saturate ore-forming fluids below the Au-Ag zones (boiling level) of epithermal deposits. This suggests that organic compounds may be the principal carriers of gold into final deposition ore zones. These metal-organic complexes decompose releasing gold in elemental form [23]. Besides the established biogenic origin of the hydrocarbons species in the gas-liquid inclusions of the Rodnikovoe deposit, based upon the presence of $C_7$–$C_{21}$ alkanes, alcohols, ketones, carboxylic acids and aldehydes (Figure 10) [24–28], and abiogenic contribution may be relatively assumed as a result of thermogenic polymerization processes (reduction of carbon oxides by hydrogen at high temperatures and pressures). The inorganic-thermogenic source is indicated by low molecular weight n-$C_{10}$–$C_{14}$ alkanes, which also make up the hydrocarbon gas mixture in inclusions of the Rodnikovoe deposit [29].

### 5.2. Physicochemical Constrains on the Selenium Endowment of the Ore-Forming Environment

The selenium-bearing LS epithermal Au-Ag deposits formed from hydrothermal fluids with $fO_2$ below the magnetite-hematite (MH) buffer and at $fSe_2/fS_2$ ratios < 1. These values prevent the deposition of selenide minerals. Nonetheless, a selenium-rich, relatively reduced (below the MH buffer) hydrothermal fluid may precipitate only silver selenides, e.g., aguilarite $Ag_4SSe$ and naumannite $Ag_2Se$, which is in fact observed at the Rodnikovoe deposit; no other selenide may be formed from this fluid. In hydrothermal solutions with low $fSe_2/fS_2$ or $H_2Se/H_2S$ ratios, selenium is incorporated into other mineral species as a solid solution by sulfur substitution [30] as in the Rodnikovoe deposit (Table 1). An increase in these ratios is only allowed at high pH and $fO_2$, which leads to the repeated replacement of Au–S complexes by Au–S–Se and Au–Se minerals [31]. The formation of the majority of selenide minerals, including AuSe, requires oxidizing conditions [32,33].

The composition of silver-sulfoselenides found in the Rodnikovoe deposit, $Ag_{1.97}(S_{0.96}Se_{0.07})_{1.03}$–$Ag_{2.01}S_{0.46}Se_{0.53}$ and $Ag_{2.03}(Se_{0.93}S_{0.04})_{0.97}$, is comparable with a series of synthetic silver sulfoselenides: $Ag_2S$–$Ag_2S_{0.4}Se_{0.6}$ and $Ag_2S_{0.3}Se_{0.7}$–$Ag_2Se$, which represent two solid solutions of the monoclinic acanthite and orthorhombic naumannite

series [34]. Aguilarite is an isostructural phase of the acanthite series, and it might be considered as Se-rich acanthite, with the same monoclinic crystal structure [34–36].

### 5.3. Au and Se Mode of Occurrence. Compositional Features of Au-Minerals in the Rodnikovoe Deposit

The variations observed in estimated empirical formulae compared with idealized stoichiometric ones, in natural acanthite and naumannite solid solution from the Rodnikovoe deposit: $Ag_{2-x}S_{1+x} \rightarrow Ag_{2+x}Se_{1-x}$ where x = 0.01–0.09 (Figure 7a), are far from random, although this fact was dismissed in the work of [36], by overgeneralizing numerous natural compositions. However, the authors of [37] did remark on a reduction of silver content relative to the ideal formula of minerals of the acanthite series, in rocks of the Dalnegorsk region. Gradual shifts, associated with an increase in the Ag content, along the solid solution compositional gradient towards the selenium region of the system, may be due to the expansion of the d-spacing, produced as the larger Se ion is substituted by S in the $Ag_2S$ [34]. The mechanism of progressive substitution of sulfur by selenium is presented in [38]: the gain of Se-content in $Ag_2(S,Se)$ occurs as a result of gradual de-sulfidation (i.e., sulfide degradation), when oxidizing conditions increase. Consequently, sulfur is oxidized to $SO_4^{2-}$ and removed from the system, which leads to the formation of more Se-enriched sulfoselenide and native silver.

Only two naturally occurring ternary compounds are known to exist for the Au–Ag–(S + Se) system [39]: uytenbogaardtite $Ag_3AuS_2$ and petrovskaite AgAuS. However, a review of several studies [40] showed that these two phases are rarely stoichiometric, but rather form finely-textured exsolution of a compositional range between end-members or a mixture of metastable phases. Analyzed compositions of natural Au-Ag-S compounds lie along the $Ag_2S$–$Au_2S$ line, in its middle part, including uytenbogaardtite and petrovskaite [40]. Contrastingly, our data show a different trend from that predicted above. The compositional trend, in this study, is directed towards a hypothetical high-sulfur compound $AuS_2$, which is yet not known in the Au–S system. On the other hand, uytenbogaardtite occurs under sulfur-excess conditions, with elevated Au/Ag ratios, and lack of sulfur-competing anions Sb, As, Bi, Se or Te [39]. This was recognized by the findings in the gold-uytenbogaardtite-acanthite assemblage: low activity of chalcogens, replacement of electrum by low-temperature uytenbogaardtite and acanthite in the presence of ultra-fine rims of Au-rich alloys. Some ore minerals from the Julietta deposit, reported as "uytenbogaardtite" [17], are in fact very far from its stoichiometric composition, being more consistent with the unnamed phase $Ag_2AuS$ (Figure 7c). According to [41], uytenbogaardtite and petrovskaite precipitate under very acidic conditions (oxidation zone) with participation of Au,Ag-bearing sulfides. Instead, the uytenbogaardtite of the Rodnikovoe deposit is formed owing to a decrease in temperature and in conditions of high activity of sulfur during the evolution of the ore-forming system, i.e., the replacement of Au-Ag alloys by sulfides of these elements.

Meanwhile, occurrences of Ag-sulfosalts, belonging to the pearceite-polybasite group, are quite common in the Chukchi low-sulfidation epithermal deposits, found in the far east of Russia [14]. The stoichiometric chemical formulae of pearceite and polybasite should be written as $[Ag_9CuS_4][(Ag,Cu)_6(As,Sb)_2S_7]$ and $[Ag_9CuS_4][(Ag,Cu)_6(Sb,As)_2S_7]$, respectively [18]. However, we obtained compositions of these minerals, of which some differ from those given by [18], e.g., in the presence of minor Au, which may be explained by the presence of inclusions with micro-, nano-gold particles. Furthermore, the presence of Se-rich sulfosalts, including rare species as spryite $Ag_8AsS_6$, argyrodite $Ag_8GeS_6$ and unnamed $Ag_5Cu_2S_5$, close to mckinstryite, among others (Table 7), clearly indicates the selenium speciation of the ore-forming system of the Rodnikovoe deposit.

**Table 7.** Composition of subordinate ore minerals from the Rodnikovoe deposit, wt.%.

| No. | Cu | Zn | Ge | Ag | Sb | As | Se | S | Total | Formula |
|---|---|---|---|---|---|---|---|---|---|---|
| 1 | | | | 71.59 | | 5.89 | 8.94 | 11.61 | 98.03 | $Ag_{8.18}As_{0.97}(S_{4.46}Se_{1.39})_{5.85}$ |
| 2 | | | | 70.94 | | 5.91 | 8.65 | 11.29 | 96.79 | $Ag_{8.23}As_{0.99}(S_{4.41}Se_{1.37})_{5.78}$ |
| 3 | | | 5.99 | 75.85 | | | 2.17 | 16.03 | 100.04 | $Ag_{8.03}Ge_{0.94}(S_{5.71}Se_{0.31})_{6.02}$ |
| 4 | | | | 65.54 | | 14.17 | 1.45 | 18.29 | 99.45 | $Ag_{3.07}As_{0.96}(S_{2.88}Se_{0.09})_{2.97}$ |
| 5 | | | | 60.96 | | 12.16 | 3.26 | 15.28 | 91.66 | $Ag_{3.18}As_{0.91}(S_{2.68}Se_{0.23})_{2.91}$ |
| 6 | | | | 63.87 | | 12.45 | 8.85 | 13.63 | 98.8 | $Ag_{3.20}As_{0.90}(S_{2.30}Se_{0.61})_{2.91}$ |
| 7 | 33.59 | 6.74 | | 3.39 | 28.19 | | | 23.88 | 96.01 | $Cu_{9.35}(Zn_{1.82}Ag_{0.56})_{2.38}Sb_{4.10}S_{13.18}$ |
| 8 | 15.01 | | | 62.84 | | | 2.75 | 17.45 | 98.05 | $Ag_{5.00}Cu_{2.03}(S_{4.67}Se_{0.30})_{4.97}$ |

1–2—spryite $Ag_8AsS_6$, 3—argyrodite $Ag_8GeS_6$, 4–6—proustite $Ag_3AsS_3$, 7—tetrahedrite $Cu_{10}(Zn,Ag)_2Sb_4S_{13}$, 8—unnamed $Ag_5Cu_2S_5$, close to mckinstryite $Ag_5Cu_3S_4$.

*5.4. Evolution of the Ore-Forming System*

The early-stage (1) of the ore-forming system, represented by the Ag-Au-aguilarite-acanthite assemblage, is silver-enriched: At this level, primary Au-Ag alloys of $Au_{0.49-0.54}$ compositions are replaced by lower-grade compositions ($Au_{0.20}$) and native silver ($Au_{0.09}$). The surge in the amount of silver occurs along with the replacement of native alloys by minerals of the acanthite series under conditions of higher sulfur fugacity, followed by an increase in the activity of copper. The latter event is interpreted based on the successive replacement of acanthite—aguilarite [$Ag_2(Se,S)$] grains by Jalpaite ($Ag_3CuS_2$). The next stage (2), characterized by the Au-Ag-uytenbogaardtite-acanthite assemblage, develops owing to gold-enriched solutions: primary $Au_{0.49-0.54}$ alloys are first replaced by high-fineness gold ($Au_{0.63-0.67}$), then by Au-bearing sulfides in association with naumannite. Overall, as recorded by the changes in ore associations, during the Rodnikovoe deposit formation, its geochemical evolution may be traced back as follows: the proportion of chalcopyrite within the sulfide parageneses increases gradually (Fe → Cu); micro-veins of silver begin to be replaced by micro-veins and rims of high-grade gold (Ag → Au or Au/Ag from 0.02 to 7.86); Se activity, and consequently its concentration, increases in acanthite and sulfosalts (S → Se); the proportion of polybasite increases in pearceite-polybasite solid solutions (As → Sb).

**6. Conclusions**

(1) The Rodnikovoe epithermal deposit is a classic low-sulfidation type, graded as a gold-silver formation (Au/Ag are 0.1–0.2) and characterized by selenium speciation, at the expenses of tellurium and antimony.

(2) Two alternating ore assemblages: silver-aguilarite-acanthite and gold-uytenbogaardtite-acanthite series are typomorphic of this deposit. In the former, $Ag_{0.49-0.56}Au_{0.44-0.51}$ alloys are replaced by solid solutions of the acanthite series and jalpaite; whereas in the latter— they are replaced by first uytenbogaardtite, then acanthite. The first assemblage is intergrown with secondary silver ($Ag_{0.77-0.91}$), while the second with higher grade gold ($Au_{0.63-0.67}$). The evolution of the ore-forming system occurred under conditions of decreasing temperature and an increasing sulfur activity: Fe → Cu; Ag → Au; S → Se; As → Sb.

(3) Organic compounds, represented by oxygen-free aliphatic (paraffins, olefins), cyclic (cycloalkanes and cycloalkenes, arenes, polycyclic aromatic hydrocarbons), oxygenated (alcohols, ethers, furans, aldehydes, ketones, carboxylic acids), nitrogenated and sulfonated hydrocarbons mixtures (1 rel.%), found in numerous fluid-gas inclusions, played an important role in the transport and accumulation of ore metals along with selenium. The compositions of these hydrocarbons are compatible with both biogenic and thermogenic origin.

(4) Rodnikovoe LS Au-Ag epithermal deposits are derived from hydrothermal fluids with $fO_2$ below the magnetite-hematite buffer, at $fSe_2/fS_2$ ratios < 1. These conditions

constrain the deposition of selenide minerals, except for the naumannite and acanthite series, and allow an active influx of selenium into sulfosalts.

(5) Most relevant typomorphic features of ore minerals: The solid solution of the acanthite series moves away from the $Ag_2S$-$Ag_2Se$ trend: $Ag_{2-x}S_{1+x} \rightarrow Ag_{2+x}Se_{1-x}$; uytenbogaardtite shows a tendency directed towards a hypothetical phase $AuS_2$, rather than a mixture with petrovskaite as expected; pearceite-polybasite minerals are mostly stoichiometric, with a subtle deviation towards Ag.

**Author Contributions:** Conceptualization, methodology, writing—review and editing, validation, N.T.; ore sampling, preparation and description, M.S.; microthermometric studies, Raman and GC-MS methods, E.S.; sampling and general geology in the field, D.B. All authors have read and agreed to the published version of the manuscript.

**Funding:** The research was carried out with the financial support of the project of the Russian Federation represented by the Ministry of Science and Higher Education of the Russian Federation No. 13.1902.21.0018 (agreement 075-15-2020-802).

**Data Availability Statement:** Not applicable.

**Acknowledgments:** We thank analysts M. Khlestov for carrying out analytical procedures and providing quantitative EMF analyses, and N. Belkina for technical assistance in preparing the manuscript. We are grateful to J. Garcia for English editing of manuscript.

**Conflicts of Interest:** The authors declare no conflict of interest. The funders had no role in the design of the study; in the collection, analyses, or interpretation of data; in the writing of the manuscript; or in the decision to publish the results.

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
