# Peer review of "The Role of Selenium and Hydrocarbons in Au-Ag Ore Formation in the Rodnikovoe Low-Sulfidation (LS) Epithermal Deposit, Kamchatka Peninsula, Russia"

_minerals, doi:10.3390/min12111418_

Round 1

Reviewer 1 Report

The submitted well written manuscript presents very interesting scientific problematics related to origin of Au-Ag mineralization in hte Rodnikovoe low-sulfidation epithermal deposit. The paper presents interesting new data and informations for mineralogy and economic geology. It should be therefore accepted and published after only a few minor revision.

The comments are as follows:

In the all text use approved nomenclature for tetrahedrite group minerals - Biagioni, C., George, L.G., Cook, N.J., Makovicky, E., Moëlo, Y., Pasero, M., Sejkora, J., Stanley, C.J., Welch, M.D. and Bosi, F. (2020) The tetrahedrite group: Nomenclature and classification. American Mineralogist, 105, 109-122.

Not use similar names as Ag,Zn-tetrahedrite but tetrahedrite-(Zn), argenotetrahedrite-(Zn) etc.

line 33 - change Figure 1 to (Figure 1)

line 56 - change Figure 2a to (Figure 2a)

line 66 - change Figure 2a to (Figure 2a)

line 68 - change Figure 2b to (Figure 2b)

line 100 - change Pearceite to pearceite

line 103 - change Table 2 to (Table 2)

line 121 - change Table 2 to (Table 2)

line 128 - change Figure 3 to (Figure 3)

line 132-134 - use in Figure 3 different symbols for data from two main assemblages

line 136 - change Figure 4 a,b,d to (Figure 4 a,b,d)

line 138 - change Figure 4 a-d to (Figure 4 a-d)

line 139-140 - change Figure 4g,h to (Figure 4g,h)

line 142 - change Figure 4 c,e,f to (Figure 4 c,e,f)

line 143 - change Figure 4 i to (Figure 4 i)

line 145 - change Figure 6 a,c,d,e  to (Figure 6 a,c,d,e)

line 146 - change Figure 5 a-d to (Figure 5 a-d)

line 146 - change Figure 5 c,d,f to (Figure 5 c,d,f)

line 147 - change Figure 5 b,e to (Figure 5 b,e)

line 171 - change Ag2Se to Ag2Se

line 172 - change Table 4 to (Table 4)

line 174 - change Figure 6a to (Figure 6a)

line 175-176 - change Table 5 to (Table 5)

line 178 - change Figure 6b to (Figure 6b)

line 187 - change Table 6 to (Table 6)

line 188 - change Figure 7a to (Figure 7a)

line 190 - change Figure 7b to (Figure 7b)

line 191-194 Figure 7 - for presentation of polybasite/pearceite use another graph - e.g. according Fig. 6 in Bindi, L., Evain, M., Spry, P. G., & Menchetti, S. (2007). The pearceite-polybasite group of minerals: Crystal chemistry and new nomenclature rules. American Mineralogist92(5-6), 918-925.

line 196 - change Table 4 to (Table 4)

line 198 - change Figure 8а to (Figure 8а)

line 200 - change Тable 5 to (Тable 5)

line 200 - change Figure 8b to (Figure 8b)

line 202 - change Figure 8c to (Figure 8c)

line 203-205 Figure 8 - quality of this figure is very poor; at part c change "pearcelite" and "polibasite" to pearceite and polybasite

line 222 - change Figure 9a to (Figure 9a)

line 226 - change Figure 9b to (Figure 9b)

line 237 - Kongsberg is a completely different genetic type of mineralization

line 244 - reference " Takahashi et al., 2002" is not according to the rules of journal

line 248-249 - correct subscripts and superscripts

line 265 - change Figure 9 to (Figure 9)

line 275 - change Ag4SSe to Ag4SSe

line 290 - 1-x to subscript

line 209 - change Figure 6a to (Figure 6a)

line 317 - change Figure 6c to (Figure 6c)

line 327 - reference "(Bindi et al., 2007)" is not according to the rules of journal

line 331 - change Table 7 to (Table 7)

line 422 - change to ..Drábek, M.; Škoda, R.

line 422 - change  (Ag2S)–naumannite (Ag2Se) to  (Ag2S)–naumannite (Ag2Se)

line 461-165 - make subscripts in chemical formulas...

Reviewer 2 Report

please see the attachment0

Reviewer 3 Report

The paper is of interest primarily because selenide occurrences are not common, and therefore worthy of description. However, there are several errors or aspects that need clarification.

L 33              (Figure 1) should be in brackets (and all subsequent figures).

Line 52, 53    volcano

L 58              argillites

L 66              have

L69               Figure 2: there is a Russian word in “Barren quartz … veins”

Table 2: Acanthite should be added to Au-Ag-uytenbogaadtite-acanthite assemblage, in addition to naumannite (note correct spelling of naumannite).

Tetrahedrite nomenclature and formulae appear incorrect (according to recent IMA definitions). Is there any compositional difference between the two?

Ag,Zn-tetrahedrite   Ag6(Cu,Ag,Zn)4(Sb,As)4S13

          Should be     argentotetrahedrite-Zn       Ag6(Cu4Zn2)Sb4S13

Ag,Zn-tetrahedrite   Cu8Ag6(Zn,Fe)2(Sb,As)4S13

          Should be ?  as above ?, argentotetrahedrite-Zn          Ag6(Cu4Zn2)Sb4S13

(Pb,Au)S       Au-bearing galena

The formula seems to imply galena with Au in solid solution. I’m not aware that galena can contain any significant level of Au solid solution, nor am I aware of any galena analyses in the literature showing gold. Do you mean galena with inclusions of native gold?

L141   Equation should be 2Ag2(S,Se)?

Figure 5: L166         Cu3SbS3 is skinnerite, not tetrahedrite

Table 6: Pearceite – polybasite analyses containing gold (up to 12.2 wt%; L187) should be flagged as possibly contaminated by micro-inclusions of native gold (as stated later in discussion, L329). This also applies to acanthite (Table 4) although the gold concentrations are lower.

Figure 6b – yellow squares almost invisible on my copy.

Figure 8b and 8c – the data in these two plots does not match that of Table 5 and 6. I did not check Figure 8a.

Figure 8b uytenbogaardite incorrect spelling. Figure 8c pearcelite incorrect spelling; polibasite incorrect spelling.
